# Does price regulation affect atorvastatin sales in India? An impact assessment through interrupted time series analysis

Sakthivel Selvaraj,[1] Habib Hasan Farooqui,[2] Aashna Mehta[1]

[1]Health Economics and Financing, Public Health Foundation of India, New Delhi, Delhi, India
[2]Indian Institute of Public Health - Delhi, Public Health Foundation of India, New Delhi, Delhi, India

**Correspondence to**
Dr Habib Hasan Farooqui;
drhabibhasan@gmail.com

## ABSTRACT

**Objective** The objective of this study was to examine the impact of medicines price regulation (Drug Price Control Order, 2013) on the market share of atorvastatin in the Indian retail market for statins.

**Setting** All Indian states, January 2012 to December 2015.

**Design** Quasi-experimental—interrupted time series analysis.

**Data** Pharmaceutical sales audit data set from IMS Health (now IQVIA) for the 48-month period from January 2012 to December 2015.

**Outcome measure** Share of atorvastatin (in percentage) in the Indian market for statins in terms of sales volumes.

**Results** We observed that the price regulation notification (Drug Price Control Orders, 2013) was associated with 0.12% (p<0.001; 95% CI 0.06 to 0.18) increase in the trend of the average monthly market share of atorvastatin (5 mg and 10 mg). After 31 months of price ceilings notification, the average market share of atorvastatin was 3.41% higher than would have been expected had the price ceilings not been notified. In sensitivity analysis, with a control, our findings remain robust, we observed a 0.16% (p<0.001; 95% CI 0.08 to 0.24) rise in the trend of average monthly market share of atorvastatin (5 mg and 10 mg) as compared with the change in the control.

**Conclusions** Price control as a public intervention did improve the relative sales of atorvastatin in the statin market in India.

## BACKGROUND

Medicines, vaccines and other medical supplies remain critical and vital elements of the health system.[1] Apart from being part of one of the vital declaration (Alma Ata Declaration, 1978), access to medicines is also one of the six targets of Millennium Development Goal 8 to develop a global partnership for development and continues to be a part of the current Sustainable Development Goals, 2030.[2] Despite being the 'pharmacy of the global south', a significant share of India's population does not have access to essential medicines.[3] A multicountry study using standard WHO methodology has reported that the median availability of a basket of 30 essential medicines in public health facilities

### Strengths and limitations of this study

► To the best of our knowledge, this is the first study to report impact of price regulation (Drug Price Control Orders, 2013) on atorvastatin sales in the Indian retail market.
► The use of quasi-experimental research design—interrupted time series analysis with a control is the key strength of this research.
► Through the use of nationally representative time series data set for a 4-year period, the study was able to show long-term impact of price regulation.
► We have not analysed the impact of price regulation on utilisation of atorvastatin in the public sector.

ranged from 0% to 30% in six Indian states during 2007.[4] While physical availability of essential medicines in health facilities is a critical indicator, the access framework emphasises the need for examining affordability besides availability. This is particularly important in free market economies where availability of medicines may not be an issue but affordability of medicines is, because it translates into poor or no access for people with low purchasing power. In India, although reported availability of cardiovascular medicines was 81% and 89% in rural and urban private pharmacies, respectively, only 59% of households were able to afford these cardiovascular medicines.[5] Literature suggests that households in India often spend a large share of their out-of-pocket (OOP) expenditure on healthcare.[6] Studies have also reported impoverishment of households because of OOP medicine expenditures.[5 7] To improve access to healthcare services and reduce financial hardship, national governments resort to various policy instruments, some of them include universal health coverage, publicly funded insurance schemes and pharmaceutical pricing policies.[8] Pharmaceutical pricing policies include policy measures to control medicine prices. These include price control measures like external reference pricing,

cost-plus price setting, measures to control add-on costs in the supply chain (such as controlling wholesaler and dispenser mark-ups) and exemption from taxes and/or tariffs.[9] Governments also use price control or limit pharmaceutical profitability as public interventions. Some Organisation for Economic Cooperation and Development countries, for instance, require manufacturers to limit prices in exchange for the subsidies they receive.[10]

India has had progressive drug pricing policies since the year 1979, through the Drug Price Control Orders (DPCO). However, the remit of DPCO had significantly reversed from the 1970s when 90% of the market was under price regulation to 2013 when only about 10% of the market was regulated.[11] The Department of Pharmaceuticals, Ministry of Chemicals and Fertilisers on 7 December 2012 notified National Pharmaceutical Pricing Policy (NPPP), 2012[12] with the objective 'to put in place a regulatory framework for pricing of drugs so as to ensure availability of required medicines—'essential medicines'—at reasonable prices even while providing sufficient opportunity for innovation and competition to support the growth of industry, thereby meeting the goals of employment and shared economic well-being of all.' The NPPP, 2012 laid down three criteria for price control: (1) regulation of prices based on 'essentiality of drugs' (ie, formulations as listed under the National List of Essential Medicines (NLEM) notified by the Ministry of Health and Family Welfare in 2011),[13] (2) control of formulation prices only and (3) market-based pricing. The DPCO, 2013 was notified on 15 May 2013 for the implementation of the NPPP, 2012.

One of the key distinguishing features of the DPCO, 2013 from the earlier avatars of DPCO include implementation of a new mechanism for price control, called market-based pricing (MBP).[14] The central idea of MBP was to compute the ceiling price of a specific formulation under the NLEM by taking a simple average of the prices to retailers of all brands, with a market share equal to or greater than 1% of the overall market for the said formulation and adding a retailer's margin of 16% to obtain the maximum retail price (MRP). In earlier DPCOs of 1979, 1986 and 1995, the mechanism of cost-plus-based pricing was followed for fixing ceiling prices of formulations by taking into account the raw material cost, conversion cost, packaging material cost and the packing charges. A maximum allowable postmanufacturing expenses was allowed over and above these costs, to the extent of 100%.[15] In addition, unlike earlier price control regimes, the new policy sought to apply price capping only on formulations (finished products) rather than the active pharmaceutical ingredient. The total number of medicines under price control as outlined in the NLEM, 2011 was 348 which translated into 628 unique strengths and dosage forms of medicines under DPCO, 2013 (for an indicative calculation of pharmaceutical price cap through DPCO, 2013, see online supplementary material). Empirical literature

suggests that while direct price control policies are effective in reducing prices and are able to control expenditures, they are unable to reduce medicine expenditures in the long run as manufacturers are able to find ways to increase sales of medicine formulations outside price control.[10] In the Indian context, the selective coverage of the policy led to concerns regarding the shift of sales from price-controlled medicines to those outside price control but within the same class of medicines as a result of change in marketing priorities of the companies who would have an incentive to market medicines outside price control.

Statins are the medications of choice for cardiovascular risk reduction (ie, prevention of heart disease, heart attack and stroke).[16] Previous research on statins use in India has highlighted that inspite of numerous statin products available in Indian retail market, only a fraction of those eligible for a statin appeared to receive the therapy. Authors reported that per capita prescribing rates for statins in India are 20 times lower than those in the USA and Canada.[16] Some of the reasons for low utilisation rate of statins in India include limited access to healthcare and affordability. We considered atorvastatin—a class of medicines prescribed to lower blood cholesterol—for our analysis because India has the highest burden of cardiovascular diseases among developing nations,[17] and among all statins, atorvastatin was the only statin under the NLEM, 2011. The price ceilings for atorvastatin (5 mg and 10 mg) were notified on 14 and 21 June 2013, respectively. The remaining strengths of atorvastatin (20 mg and 40 mg), other statins such as rosuvastatin, simvastatin and all combinations of atorvastatin (eg, atorvastatin+acetylsalicylic acid) remain outside price regulation (for detailed information on statin market, see tables S1, S2, and S3 in online supplementary material).

The effectiveness of pharmaceutical pricing policies depends on the scope and mechanisms around which the regulation is designed. It has been argued that partial price regulation involving one particular drug formulation without considering other equivalent therapeutic formulation may result in switch away from price-regulated products to unregulated pharmaceutical products.[18] We hypothesised that on account of new price regulation (DPCO, 2013) there would be a shift in sales of statins from price-controlled atorvastatin towards therapeutically equivalent non-price-controlled statins. The key objective of this study was to examine the impact of medicine price regulation (Drug Price Control Order, 2013) on the market share of atorvastatin in the Indian retail market for statins through interrupted time series analysis—a quasi-experimental design. To the best of our knowledge, our study is the first attempt from India to generate evidence on the impact of price regulation (DPCO, 2013) on the utilisation of atorvastatin used in cardiovascular diseases.

## MATERIALS AND METHODS

### Data

We used the sales audit data set from IMS Health (now IQVIA) of the Indian pharmaceutical market for a 48-month period from January 2012 to December 2015 for this study. IMS Health (now IQVIA) is a for-profit organisation that collects and provides data and information on pharmaceutical market intelligence in over 100 countries around the world. The Indian pharmaceutical sales data are collected from a panel of 5600 stockists across different regions of the country and extrapolated to reflect the private sector sales in the entire country. This comprises the sales made by the stockists to the retailers, hospitals as well as dispensing doctors. The data organise pharmaceuticals based on anatomical therapeutic classification of the European Pharmaceutical Market Research Association. We used this information to identify the private market for statins in the country. The data do not capture the public sector utilisation of medicines and, therefore, our analysis pertains only to the impact of the policy on the private sector utilisation of statins in the country.

### Outcomes

Our primary outcome measure was market share (in percentage) in terms of sales volumes. Sales volumes are provided in the sales audit data set in term of standard units (SUs) which are defined by IMS Health (now IQVIA) as the smallest dose of formulation which can be one tablet or capsule for oral solids, one phial or ampoule for injectables and so on. We computed the monthly share of atorvastatin (5 mg and 10 mg), the statin under the NLEM and hence within the ambit of price regulation, in terms of sales volume in the statins market for the time period under study. Medicine sales were taken as a proxy for utilisation of the specific formulation for the purpose of our analysis.

### Research design

We used interrupted time series, a quasi-experimental research design to capture the impact of price ceiling on utilisation of atorvastatin (5 mg and 10 mg were notified under DPCO in June 2013). Rosuvastatin (all strengths), a statin outside price regulation, was used as the control to further strengthen our research design.

### Statistical analysis

The intervention under study is the notification of price ceilings for atorvastatin (5 mg and 10 mg). Specifically, the price ceiling for atorvastatin 5 mg and atorvastatin 10 mg were notified on 14 June 2013 and 21 June 2013, respectively. The 48-month period under study was distributed into two segments, preintervention period of 17 months, from January 2012 to May 2013 and postintervention period of 31 months, from June 2013 to December 2015. We did not take into consideration the period after December 2015 as a new NLEM was notified in this period. Based on the new NLEM, 2015, price

ceilings were subsequently notified for atorvastatin 10 mg, 20 mg and 40 mg in 2016.

We used segmented linear regression analysis to detect the preintervention trends, postintervention level changes and changes in postintervention trends relative to the preintervention trends in the use of atorvastatin (5 mg and 10 mg). The dependent variable ($Y_t$) was 'market share' of atorvastatin (5 mg and 10 mg) in terms of sales volumes while 'time' appeared as an independent variable. We fitted a least square regression line to the two segments of the continuous variable time. In addition, we introduced two binary variables to estimate immediate level change after the intervention (variable name: intervention) as well as trend change (variable name: time after intervention) in the market share of regulated atorvastatin (see equation 1). The variable 'intervention' was 0 for the preintervention period and 1 for the postintervention period starting June 2013 (model 1). Time after intervention was a continuous variable starting June 2013.

$Y_t = \alpha + \beta_1 \text{ time}_t + \beta_2 \text{ intervention}_t + \beta_3 \text{ time after intervention}_t + \varepsilon_t \ldots$equation 1

The segmented regression helped us statistically estimate the change in the intercept and the slope coefficients between the preintervention and postintervention period. We checked the model for autocorrelation with the help of Durbin-Watson statistic, autocorrelation (ac) and partial autocorrelation (pac) estimates and plots of the residuals. We did not detect autocorrelation in our models (see figures 1a, b, c, 2a, b and c in online supplementary material for details). Seasonality is unlikely to influence the outcome measure as the group of medicines under study is intended for a chronic condition and meant to be consumed throughout the lifetime of the patients.

In addition, we ran an alternate model (model 2) wherein we excluded the adjustment period of 3 months from June to August 2013 since manufacturers were provided a maximum of 45 days to adjust their maximum retail prices in accordance with the notified ceiling price. Inspecting the data also revealed that this period saw a steep decline in the share of price regulated atorvastatin in the statin market which suggests that the suppliers were withdrawing the stocks from the market to relabel the medicine packs with the revised MRPs. However, we do not have a way of knowing which manufacturers or how many of them adjusted the ceiling prices in a weeks' time, 2 weeks' time, 3 weeks' time and so on. Further, we do not believe that the availability of this information would have impacted our results significantly or that the implementation period was long enough to affect our results. A counterfactual was introduced into both the models, that is, trend in utilisation of atorvastatin in the postintervention period had the price ceilings not been notified. It is predicted that in the absence of price ceiling, preintervention trend in utilisation would have remained constant in the postintervention period. We computed the difference between the predicted values of the market share of atorvastatin for the actual as well

as the counterfactual scenarios to get the estimate of the absolute policy effect.

Finally, a control was introduced in the model (model 3) to strengthen our study design. The market share of rosuvastatin, a statin completely outside the ambit of price regulation, was used as control. Rosuvastatin was chosen as control instead of other statins such as simvastatin, because rosuvastatin is the second highest-selling plain formulation in both value and volume terms after atorvastatin. The intention was to control for time-varying confounders and other interventions that may have affected the outcome of interest. We computed the difference in the respective market shares of atorvastatin (5 mg and 10 mg) under price regulation and rosuvastatin, our comparison group. We then ran the above-described interrupted time series model with the difference in the market share as our new dependent variable. This method had the advantage of simultaneously controlling the preintervention differences in the utilisation level and trend between the two statins. Once again, two separate models were run—with and without the adjustment period (model 3). We, additionally ran interrupted time series model on total sales volumes (in SUs) per capita instead of market share for atorvastatin (5 mg and 10 mg) as well statins as a whole to check if the policy has an impact on the sales volumes of the entire statins market.

### Patient and public involvement

Patients or the public were not involved in this study, which is based on secondary data.

### RESULTS

Our analysis suggests that statin market which was worth INR 22.90 billion in 2015, accounted for 2.25% of the Indian pharmaceutical market in terms of sales value and 0.63% of the market in terms of sales volume (table 1). The market value and volume of statins increased between 2012 and 2015 both in absolute terms as well as in terms of relative share in the overall pharmaceutical market. It is interesting to note that for the time period 2012 to 2015, the share of fixed-dose combinations of statins have been rising and that of plain formulation has been falling in terms of both sales values and volumes.

The contribution of atorvastatin in overall statin market was 44.9% and 48.9% in volume and value terms in the year 2012, respectively, which went down to 35.7% and 38.2% of the overall statin market by the year 2015 (table 2). We observed that the market share of regulated atorvastatin (5 mg and 10 mg) in the statins market has also been falling in terms of both sales values and volumes for the period 2012 to 2015. For the same period, the market share of another statin, rosuvastatin, has been increasing in both value and volume terms. The difference between the market shares of both the statins in terms of sales volume has been narrowing over the study period. In terms of sales values, however, the difference

**Table 1**  Statins in the Indian pharmaceutical market

| Statins market | Volumes in billion SUs (%) | | | | Values in INR billions (%) | | | |
|---|---|---|---|---|---|---|---|---|
| | 2012 | 2013 | 2014 | 2015 | 2012 | 2013 | 2014 | 2015 |
| Overall statins market (share in the Indian pharmaceutical market) | 2.50 (0.50) | 2.81 (0.56) | 3.15 (0.60) | 3.56 (0.63) | 15.47 (2.16) | 16.92 (2.14) | 19.41 (2.20) | 22.90 (2.25) |
| Fixed Dose Combinations (share in statins market) | 0.94 (37.56) | 1.14 (40.44) | 1.31 (41.76) | 1.55 (43.40) | 4.56 (29.46) | 5.21 (30.79) | 6.48 (33.40) | 7.92 (34.56) |
| Plain formulations (share in statins market) | 1.56 (62.44) | 1.67 (59.56) | 1.83 (58.24) | 2.02 (56.60) | 10.91 (70.54) | 11.71 (69.21) | 12.93 (66.60) | 14.99 (65.44) |
| Total market | 504 357.87 | 504 757.49 | 523 098.93 | 563 619.26 | 715 163.62 | 788 943.16 | 882 423.34 | 1 015 741.19 |

INR refers to Indian national rupee.
SUs, standard units.

**Table 2**  Share of atorvastatin 5 mg and 10 mg, other strengths of atorvastatin and rosuvastatin in the statins market

| | Sales volumes (%) | | | | Sales values (%) | | | |
|---|---|---|---|---|---|---|---|---|
| | **2012** | **2013** | **2014** | **2015** | **2012** | **2013** | **2014** | **2015** |
| (A) Atorvastatin (5 mg and 10 mg) (%) | 32.34 | 28.87 | 26.29 | 24.1 | 25.84 | 22.4 | 17.01 | 15.3 |
| (B) Atorvastatin (other strengths) (%) | 12.63 | 11.88 | 11.81 | 11.64 | 23.09 | 22.57 | 23.19 | 22.93 |
| (C=A+B) Atorvastatin (all strengths) | 44.97 | 40.75 | 38.1 | 35.73 | 48.93 | 44.96 | 40.2 | 38.23 |
| (D) Rosuvastatin (all strength) (%) | 15.79 | 17.53 | 19.15 | 20.12 | 19.65 | 22.66 | 25.16 | 26.27 |
| (E=A- D) Difference between atorvastatin (5 mg and 10 mg) and rosuvastatin (%) | 16.55 | 11.34 | 7.14 | 3.98 | 6.19 | −0.26 | −8.15 | −10.97 |

between the two market shares turned negative in 2013 and the gap has been widening up to 2015.

Our results from segmented regression analysis (model 1) suggest that postintervention, there was an immediate reduction in the average monthly market share of atorvastatin by 0.25% but this change in level was insignificant. This was followed by a significant increase in the trend of 0.12% (p<0.001) per month. Figure 1A demonstrates fitted values (solid line) of the market share of atorvastatin in the preintervention and postintervention period. Additionally, the figure also captures the counterfactual fitted values of market share (dotted line). The counterfactual represents the scenario had the price ceilings for atorvastatin not been notified during the period under study. The vertical line in the figure at June 2013 represents the intervention period which in our case is the month in which price ceilings were notified for atorvastatin (5 mg and 10 mg). It is evident from figure 1A and table 3 that the effect was negative for the first couple of months. This was followed by a sustained positive effect for the remaining duration. In December 2015, the 31st month after the notification of price ceilings, the average market share of atorvastatin was 3.41% higher than would have been expected had the price ceilings not been notified.

We ran another model to check whether the implementation period impacted our results significantly. In model 2, we excluded the period from June 2013 to August 2013. On excluding the 3-month period, we did not observe a significant modification in our results (see figure 1B). The postintervention level change continued to be negative and insignificant, although the magnitude increased by 0.23 and the postintervention trend change slightly reduced by 0.01 but continued to be significant (p<0.001). In December 2015, the 31st month after the notification of price ceilings, the average market share of atorvastatin was 2.28% higher than would have been expected had the price ceilings not been notified (see table S4 in online supplementary material for absolute policy effect from the intervention point June 2013 up to December 2015). Excluding the implementation period did not alter our results significantly.

To increase the robustness of our analysis, we introduced a control (model 3), another statin, rosuvastatin (all strengths), which was outside the ambit of price regulation. Table 4 and figure 2 present result of the segmented regression analysis with the difference in the market shares of atorvastatin (5 mg and 10 mg) and rosuvastatin as the dependent variable. We observed a 0.69% (p>0.05) immediate postintervention drop in the

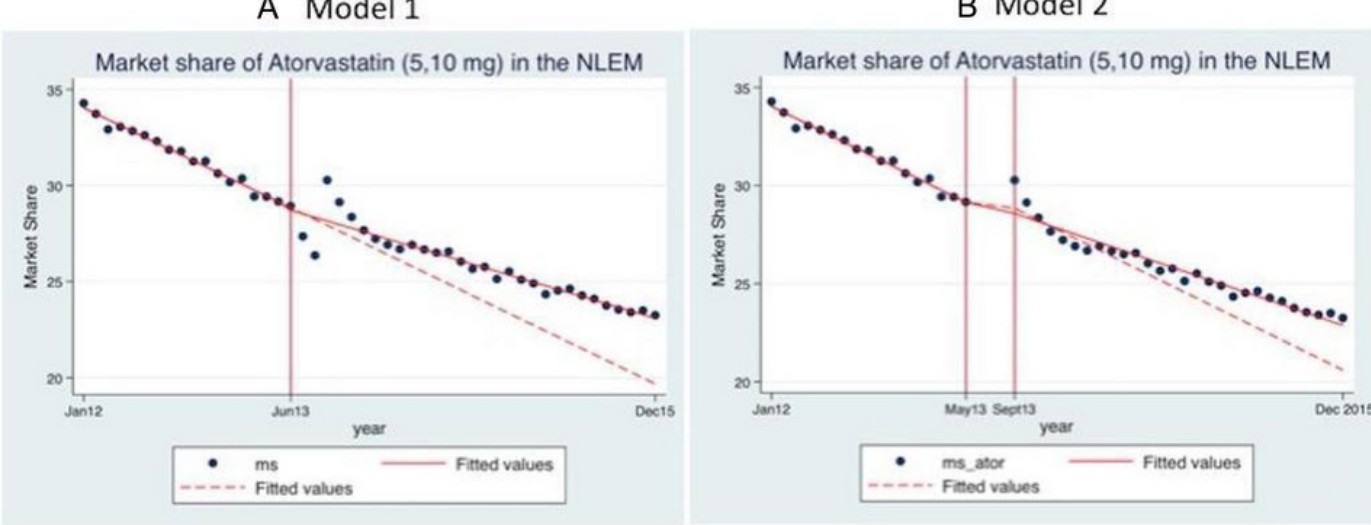

**Figure 1**  Fitted values of the market share of atorvastatin 5 mg and 10 mg, model 1 and model 2—actual and counterfactual. ms, market share of atorvastatin (5 mg and 10 mg).

**Table 3** Segmented regression analysis results for drug utilisation (atorvastatin 5 mg and 10 mg)

| Variable | Model 1 | | | Model 2 | | |
|---|---|---|---|---|---|---|
| | Coefficients | 95% CI | | Coefficients | 95% CI | |
| Time | −0.31*** | −0.36 | −0.25 | −0.31*** | −0.36 | −0.25 |
| Intervention (level change) | −0.25 | −0.9 | 0.41 | −0.48 | −1.12 | 0.15 |
| Time after intervention (trend change) | 0.12*** | 0.06 | 0.18 | 0.11*** | 0.05 | 0.17 |
| _Const | 34.34*** | 33.78 | 34.9 | 34.34*** | 33.8 | 34.88 |
| Number of observations | 48 (preintervention: 17 and postintervention: 31) | | | 45 (preintervention: 17 and postintervention: 28) | | |
| $R^2$ | 0.9828 | | | 0.9763 | | |

***P<0.001.
**P<0.05.

average monthly difference in the market share of atorvastatin (5 mg and 10 mg) followed by a 0.16% (p<0.001) rise in the monthly trend as compared with the change in the control, rosuvastatin. Our results after factoring in the control reiterate robustness of findings, that is, the average monthly trend of the difference in the market share of atorvastatin (5 mg and 10 mg) is positive in the postintervention period. Therefore, our null hypothesis that price regulation would lead to a switch away from price-regulated atorvastatin (5 mg and 10 mg) towards statins outside price ceiling stands rejected.

In summary, the DPCO, 2013 regulation led to a shift towards the price-regulated atorvastatin (5 mg and 10 mg) from other statins which were not under price regulation. Finally, we did not observe any effect of the policy on the per capita sales volumes (in SUs) of either atorvastatin (5 mg and 10 mg) or statins as a whole. The policy did not significantly impact the sales of regulated atorvastatin in absolute terms or statins as a whole. The impact was relative and therefore can be interpreted as a switch towards the regulated atorvastatin.

## DISCUSSION
We have generated robust evidence on the impact of price regulation policy (DPCO, 2013) on the sales of

**Table 4** Segmented regression analysis results for drug utilisation (atorvastatin 5 mg and 10 mg with control)

| Variable | Model 3 | | |
|---|---|---|---|
| | Coefficients | 95% CI | |
| Time | −0.45*** | −0.52 | −0.37 |
| Intervention (level change) | −0.69 | −1.54 | 0.16 |
| Time after intervention (trend change) | 0.16*** | 0.08 | 0.24 |
| _Const | 19.49*** | 18.76 | 20.22 |
| Number of observations | 48 (preintervention: 17 and postintervention: 31) | | |
| $R^2$ | 0.9808 | | |

**P<0.05.
***P<0.001.

atorvastatin (5 mg and 10 mg) through the use of interrupted time series analysis—a quasi-experimental design. The use of interrupted time series analysis to measure the policy impact by comparing preintervention versus postintervention trends has increased in recent past especially to assess impact of health-related interventions.[19–25] Our research design, analytic approach and reporting conform to well-established methodological standards.[26 27]

We observed that the baseline trend for average monthly atorvastatin sales was declining during the study period (January 2012 to December 2015). Some of the reasons for such trend includes prescription preferences, availability of therapeutically substitutable products and marketing practices of manufacturers. However, the sales of the price-controlled formulation of atorvastatin (5 mg and 10 mg) relative to other statins increased after the ceiling price notification in June 2013. Further, after an initial downward shift in sales, which could have resulted from the recall of existing packs of atorvastatin from the

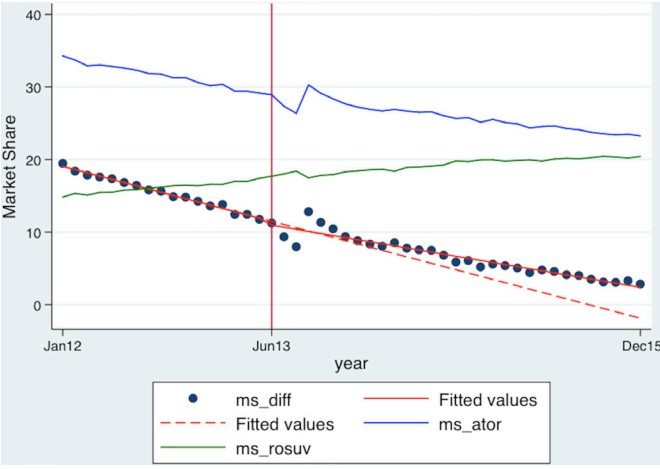

**Figure 2** Fitted values of the difference in the market share of atorvastatin (5 mg, 10 mg) and rosuvastatin (all strengths)—actual and counterfactual. ms_ator, the market share of atorvastatin (5 mg and 10 mg); ms_diff, the difference between the market share of atorvastatin (5 mg and 10 mg) and rosuvastatin (all strength); ms_rosuv, the market share of rosuvastatin (all strengths).

retail market for repackaging and reprinting the new MRPs[28] as mandated in DPCO, 2013, there was a sustained positive effect of regulation as reflected in increasing monthly sales. This sustained positive effect may have resulted primarily because of price reductions leading to increased utilisation of the price-regulated atorvastatin (5 mg and 10 mg) in comparison to other statins on account of increased affordability. Similarly, Damiani *et al* evaluated national and regional cost containment measures (ie, revised national reimbursement criteria and regional copayment) on statins use in Italy using interrupted time series analysis and reported that the regional copayment was associated with a small increase in trend of statin use, whereas restriction to reimbursement interventions was associated with an immediate drop and a decrease in trend of statin use, highlighting that public policies directed towards cost containment may impact statin utilisation.[22]

Empirical literature suggests that direct price control policies are usually unable to reduce medicine expenditures as manufacturers are able to find ways to increase sales of medicine formulations outside regulation.[10] Recent research on price control of antihypertensive medicines from Korea reported some unintended effect of the policy, that is, drug price reduction resulted in drug overutilisation and use of prohibited combinations. Also, the utilisation of drugs, which was not affected by the drug price reduction, increased by 12.3%.[29] Studies from Norway and Finland have reported that policy interventions directed towards physicians to prescribe less expensive statins and restricting reimbursement of expensive statins have resulted in increase in the consumption of statins and, simultaneously, a decrease in the expenditure.[30 31] WHO guidelines on pharmaceutical pricing policies also suggest that at any given point of time national governments should employ a judicious mix of different policy instruments to control medicine prices and expenditure.[9]

Our study has certain limitations. We have evaluated the impact of price regulation on only statins market; our findings are not representative for other medicines or formulations under price control. In addition, after the announcement of NLEM, 2015, few more strengths of atorvastatin (20 mg and 40 mg) have been brought under price control through Drugs (Price Control) Amendment Order, 2016.[32] To ascertain the effect of increased coverage of price regulation, another segmented regression analysis should be conducted after sufficient time period has lapsed after implementation of Drugs (Price Control) Amendment Order, 2016. More research is needed to measure the impact of price regulation in other markets on the basis of other medicines which are part of NLEM, 2015. We have also not evaluated the effect of increased utilisation of atorvastatin on health outcomes. We have not analysed the impact of price regulation on atorvastatin utilisation in the public sector. Although the price regulation policy has no intended effect on public sector utilisation since medicine procurement in public sector is handled through medical services corporations which procure medicines through tender-based system.

## CONCLUSIONS

The evidence emerging from our study indicates that the medicine price regulation (DPCO, 2013) had sustained positive impact on atorvastatin utilisation in comparison to other statins as reflected in increasing average monthly sales in the postintervention period. However, the policy impact was relative, interpreted as a switch towards the regulated atorvastatin. It should be noted that the impact of price regulation may vary across other medicines or formulation on account of competition and concentration in each therapeutic segment, manufacturer's marketing practices, physician's prescribing behaviour and consumer behaviour. In addition, policy makers should also consider unintended effects of the policy, such as manufacturers' response to reduced prices and consequently reduced profitability of the formulations. To enhance affordability of medicines, the price regulation should be accompanied with other regulatory reforms and policy measures; universal access to essential medicines could be one of them.

**Acknowledgements** The authors would like to thank Dr Anup Karan (Asscoiate Professor, Indian Institute of Public Health, Public Health Foundation of India) for his critical and constructive comments on the methodology.

**Contributors** SS, HHF and AM conceived the idea, designed the analysis, conducted data analysis and wrote the first draft of the paper. SS, HHF and AM conducted the literature review and the interpretation of the results. SS, HHF and AM revised and edited the manuscript to its final stages. All the authors approved the final manuscript version.

**Funding** HHF is supported by a Wellcome Trust Capacity Strengthening Strategic Award (084754/Z/08/A) to the Public Health Foundation of India and a consortium of UK universities.

**Competing interests** None declared.

**Patient consent for publication** Not required.

**Provenance and peer review** Not commissioned; externally peer reviewed.

**Data sharing statement** The IMS health data is available on request, at the approval of IMS Health.

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
