## [Reviewer comments · BMJ Open]

ARTICLE DETAILS

TITLE (PROVISIONAL)	Does Price Regulation Affect Atorvastatin Sales in India? An Impact Assessment Through Interrupted Time Series Analysis
AUTHORS	Selvaraj, Sakthivel; Farooqui, Habib; Mehta, Aashna

VERSION 1 – REVIEW

REVIEWER	Sonak Pastakia Purdue University/Purdue Kenya Partnership, Kenya
REVIEW RETURNED	12-Aug-2017

GENERAL COMMENTS	There are a few minor grammatical errors throughout the paper that should be edited out but these are fairly trivial issues that could quickly be addressed. i would be a bit more cautious with the conclusions since i think there are other aspects to this topic which could give additional insight. I would be interested in seeing how manufacturers changed their marketing expenditures based on this change in pricing as I'm assuming profit margins changed with the other competitors. I think that is particularly interesting finding because it illustrates how susceptible prescribers are to marketing strategies. can you also comment on the absolute numbers? Was there an overall increase in statin prescription or did that go down as well? Market share is the primary point here as you rightly describe but I'm also interested in the trend with statin prescription in general. I would discuss the findings in the supplementary appendix Table 2 to describe the trend with overall statin prescribing in the country. It looks like more statin prescriptions were given out. Do you have any explanation for that? Is it because the price was now lower or solely because of health trends. i would also make sure all abbreviations in the table and other parts of the paper are spelled out. What is Sus in table 2? Overall, I found the paper provided much needed initial insight on the important topic of price regulation and the subsequent impact on utilization. I look forward to seeing this in publication so other countries can learn from India's experience. For the editors-I'm not an economist by training so when the statistical review is performed, I would hope that an economist could confirm that appropriate economics analytical approaches were used.
---

REVIEWER	Olivier J. Wouters London School of Economics and Political Science, U.K.
REVIEW RETURNED	18-Aug-2017

GENERAL COMMENTS	BMJ Open (Manuscript #2017-018347)
---

The aim of this article was to evaluate the impact of a new pharmaceutical pricing policy in India, introduced in June 2013, on sales of the cholesterol-lowering drug atorvastatin. The authors conducted a segmented regression analysis using monthly data on sales of the drug (Jan. 2012 - Dec. 2015). They found that the policy led to a significant increase in the sales trend, which was 0.12% (95% confidence interval: 0.06%-0.18%) higher in the post-intervention period than in the pre-intervention period ($p < 0.001$). The authors estimated that the market share of atorvastatin — as a proportion of total sales of statins — was 3.41% higher 31 months after the policy change than it would have been had the policy not been introduced.

Thank you for the opportunity to review this article. I appreciate that the authors used a strong, quasi-experimental study design to answer their research question. Yet I have some comments about the study which I think would need to be addressed before the article is suitable for publication. Also, I would suggest that the authors closely follow the methodological and reporting recommendations for interrupted time series studies developed by Jandoc et al (2015) to ensure all key details are presented in the methods and results sections.

Major comments

Background — This section is long and repetitive. The key points are that the Indian government is struggling to ensure the availability and affordability of medicines in the country, and that the health ministry introduced a new pricing policy in 2013 to improve access to medicines.

- The authors could consider condensing the first three paragraphs into one and then describing the relevant parts of the new policy (Drug Price Control Order of 2013) in one or two paragraphs.
- The description of the new policy is unclear. Which 348 essential medicines were subjected to the policy, and why did you decide to focus on the market-based pricing element of the policy? Does medicine here refer to an active ingredient or a form-strength combination of an ingredient? What do you mean by the “control of formulation prices”?
 - After reading the background section, the expected (or intended) effect of the policy on market shares remains unclear. It would be helpful if you could offer a rationale in the background for why you conducted this research.
- Why did the authors select atorvastatin as their case study? The authors say that India has “the highest burden of cardiovascular diseases among developing nations”, but I thought they were interested in studying the general impact of the policy change on the market shares of medicines affected by the new regulation — rather than the specific impact on the market shares of cardiovascular products.
 - Are data available for other medicines? It would have been interesting to select a larger sample of widely-used medicines, or at least a few key

	products to improve the generalisability of the results.  The authors use many acronyms in the article, which makes it tedious to read. I would suggest that the authors remove all non-standard abbreviations which are not used at least four times in the paper and instead write out the full terms. Materials and Methods — I have comments and questions about the methods.  Which month in 2013 was the policy introduced? It appears to be June based on the first sentence under “statistical analysis”, but there needs to be a clear statement specifying when the policy came into effect.  Was this the only change that occurred around that time which might have influenced the market shares of atorvastatin? In the discussion (p. 11), you mention that “the Government of India has also implemented a host of policy intervention apart from NPPP 2012 in order to accelerate affordability of medicines and reduce out-of-pocket payments on medicines.” Could you please elaborate on this statement, since such policy interventions could threaten the internal validity of your findings? In the appendix, could you please list the p-value for the test for positive and negative autocorrelation along with the Durbin-Watson test statistic? It would be good to include a statement that seasonality is unlikely to influence the outcome measure, if this is correct; otherwise you would need to account for seasonality in your model. Could you please re-run the model to allow for a lagged treatment effect? As shown in Figure 1, there are wild data points in the first five months after the policy change. You should drop these observations if you think the fluctuations occurred during a period of adjustment to the new policy and were not due to random variation, as done in other studies (Leopold et al 2014). Otherwise you might obtain an incorrect estimate of the treatment effect (Wagner et al 2002; Jandoc et al 2015). Could you also re-run the model with total volume per month per capita as a secondary outcome measure, as done in similar studies (e.g., Leopold et al 2014; Garabedian et al 2012)? The impact of the policy change on market shares and volumes might differ. To estimate the treatment effect relative to the control group, I would suggest you use the difference in the market shares between the two medicines over time as the outcome variable, as done in other studies (Law et al 2008, Penfold and Zhang 2008). Why did you select rosuvastatin instead of simvastatin (or another statin) as the control?  As mentioned above, it is unclear which medicines were subjected to the policy. Why was atorvastatin affected by the new regulation, but not other statins like rosuvastatin and simvastatin? This information is important for determining whether rosuvastatin is an appropriate control group or not.
--	---

- The subsection on the null hypothesis is incorrect and misleading. The null hypotheses relate to each variable in your model, usually specifying that the value of a coefficient is 0, holding the other variables constant. As all the analyses were two-tailed with a type I error rate of 0.05, the model would pick up an increase or decrease in the market share of atorvastatin (not just a decrease, as you seem to suggest). Also, you are describing the alternative hypothesis in the text, not the null hypothesis.

Results — I would suggest the authors rewrite this section after considering the comments above. Please ensure you follow the reporting recommendations of Jandoc et al (2015). I have a few minor points:

- Table 1 is difficult to read. I would indent the atorvastatin strengths to make it clear that those figures add up to the total across all strengths (first row). The authors may want to somehow distinguish the last two rows from the rest (e.g., add a blank row above), since they present different types of statistics. Also, the units in the last row are billions of standard units (not % market share), so the column headings do not apply.
- I would suggest you present the trends for both the treatment and control group (rosuvastatin market shares) in Figure 1, along with the differences between the two.
- Could you please remove the sentence about the null hypothesis (p. 10) for the reasons mentioned above?

Discussion — I have concerns about the generalisability of your findings. Also, without a clear statement of purpose in the background, it is difficult to understand the policy implications of your research, especially since you only examined the impact of the policy change on the market share of a single product.

- What can health stakeholders in India and elsewhere learn from this study? Why is this important evidence for the wider health policy audience?
- A key limitation of this study is that the authors only analysed the market shares of atorvastatin, so it is difficult to generalise your results to other products (or even other strengths of atorvastatin than those included in your analysis). You may wish to discuss why these results may or may not hold for other products.
- Please say “possible reasons” instead of “notable reasons” (p. 10), since this is your own interpretation.
- Could you please discuss what your study adds to the literature?

Conclusion — I would rewrite this section to clearly and concisely communicate the key finding of your study and the wider policy implications of your results. I would not repeat points relating to the study sample, such as “Two strengths of Atorvastatin (20mg and 40mg) were included in ... left out of the list.”

Minor comments

- There are many grammatical mistakes and formatting errors in the paper, so please proof read the article again carefully. Here are some examples:

- “Regulation” is misspelled (keywords, p. 2)
 - “Another” should be capitalised (last sentence of paragraph 3, p. 4)
 - “Market-based pricing” should not be capitalised (1st paragraph p. 5)
 - Reference 17 (p. 7) should appear in the same format as the others, i.e. superscript number
 - The total sales volume was 28.9% not 28.8% (top of p. 8)
 - There is an extra closing parenthesis (p.8)
 - You’ve already defined the acronym NLEM, so there is no need to re-define it in the conclusion (p. 12)
- In the background section, the sentence “For example, the Organization for Economic Cooperation for [sic] Development (OECD) countries require manufacturers to limit prices in exchange of [sic] subsidies they receive” (p. 4-5) is factually incorrect. You could say that “some” OECD countries do this, but it is not the case in all. You should also clarify what you mean by subsidies.
 - Please note that the name of the institution is the Organization for Economic Cooperation and Development (not “for”), and the sentence should read “for subsidies” (not “of subsidies”).
 - I would not capitalize the generic names of medicines, like atorvastatin and rosuvastatin.
 - The anatomical therapeutic chemical classification system was developed by the WHO Collaborating Centre for Drug Statistics Methodology, not the European Pharmaceutical Market Research Association (p. 6), which is an industry association.
 - The name of the organisation which provided the data is now QuintilesIMS, not IMS Quantile (p. 5)
 - I would present the 95% confidence intervals for the point estimates in the abstract.

References

- Garabedian LF, Ross-Degnan D, Ratanawijitrasin S, Stephens P, Wagner AK (2012). Impact of universal health insurance coverage in Thailand on sales and market share of medicines for non-communicable diseases: an interrupted time series study. *BMJ Open*, **2**: e001686.
- Jandoc R, Burden AM, Mamdani M, Lévesque LE, Cadarette SM (2015). Interrupted time series analysis in drug utilization research is increasing: systematic review and recommendations. *Journal of Clinical Epidemiology*, **68**: 950-956.
- Law MR, Majumdar SR, Soumerai SB (2008). Effect of illicit direct to consumer advertising on use of etanercept, mometasone, and tegaserod in Canada: controlled longitudinal study. *BMJ*, **337**: a1056.
- Leopold C, Zhang F, Mantel-Teeuwisse AK, Vogler S, Valkova S, Ross-Degnan D, Wagner AK. Impact of pharmaceutical policy interventions on utilization of antipsychotic medicines in Finland and Portugal in times of economic

	recession: interrupted time series analyses. International Journal for Equity in Health, 13: 53. Penfold RB, Zhang F (2013). Use of interrupted time series analysis in evaluating health care quality improvements. Academic Pediatrics, 13(6S): S38-S44. Wagner AK, Soumerai SB, Zhang F, Ross-Degnan D (2002). Segmented regression analysis of interrupted time series studies in medication use research. Journal of Clinical Pharmacy and Therapeutics, 27: 299-309.
--	---

VERSION 1 – AUTHOR RESPONSE

Reviewer 1:

Comment 1: There are a few minor grammatical errors throughout the paper that should be edited out but these are fairly trivial issues that could quickly be addressed.

Response: We have reviewed that manuscript and made necessary corrections.

Comment 2: I would be a bit more cautious with the conclusions since i think there are other aspects to this topic which could give additional insight. I would be interested in seeing how manufacturers changed their marketing expenditures based on this change in pricing as I'm assuming profit margins changed with the other competitors. I think that is particularly interesting finding because it illustrates how susceptible prescribers are to marketing strategies.

Response: Studying the changes in the marketing expenditure in response to price change, although an interesting enquiry, is beyond the scope of the current study which is focused on the impact on utilisation. The information required to do this using interrupted time series analysis would require monthly data on marketing expenditure by all companies marketing Atorvastatin in the country. Although annual data might be available in company annual reports, monthly data would be hard to obtain. Since enough number of years haven't passed since the regulation was introduced, we will not be able to undertake time series analysis using annual data. Besides, we would not be able to gather information on medicines (exact strengths and dosage forms) toward which the expenditure is being directed.

Comment 3: can you also comment on the absolute numbers? Was there an overall increase in statin prescription or did that go down as well?

Market share is the primary point here as you rightly describe but I'm also interested in the trend with statin prescription in general. I would discuss the findings in the supplementary appendix Table 2 to describe the trend with overall statin prescribing in the country. It looks like more statin prescriptions were given out. Do you have any explanation for that? Is it because the price was now lower or solely because of health trends.

Response: The suggested changes have been incorporated in the manuscript. Table 2 in the supplementary appendix has been moved to the main text of the manuscript as table 1. It is difficult to comment on the specific reasons behind the increase in statin sales for the period under study. One of the reasons could be already high burden of cardiovascular disease and increasing awareness, leading to better health-seeking behaviour. However, since only a small proportion of the statins market is under price regulation, and the sales for the price regulated statins demonstrated a downward trend for the duration of the study, an increase in the sales of the overall segment cannot be attributed to the policy.

Comment 4: i would also make sure all abbreviations in the table and other parts of the paper are spelled out. What is Sus in table 2?

Response: we have reviewed that manuscript and made necessary modification.

Comment 5: Overall, I found the paper provided much needed initial insight on the important topic of price regulation and the subsequent impact on utilization. I look forward to seeing this in publication so other countries can learn from India's experience.

Response: Thanks. This is very encouraging.

Reviewer 2:

Background — This section is long and repetitive. The key points are that the Indian government is struggling to ensure the availability and affordability of medicines in the country, and that the health ministry introduced a new pricing policy in 2013 to improve access to medicines.

- The authors could consider condensing the first three paragraphs into one and then describing the relevant parts of the new policy (Drug Price Control Order of 2013) in one or two paragraphs.

Response: The section has been condensed, where possible. However, we deem fit to retain detailed information on access to medicines in India in the interest of uninitiated readers.

- The description of the new policy is unclear. Which 348 essential medicines were subjected to the policy, and why did you decide to focus on the market-based pricing element of the policy? Does medicine here refer to an active ingredient or a form- strength combination of an ingredient? What do you mean by the “control of formulation prices”?

Response: The following sentence in the background section of the manuscript clarifies the meaning of “control of formulation prices”. In addition, unlike earlier price control regimes, the new policy, sought to apply price capping only on formulations (finished products) rather than the Active Pharmaceutical Ingredients (API).

The section has been further modified for greater clarity, as necessary.

- o After reading the background section, the expected (or intended) effect of the policy on market shares remains unclear. It would be helpful if you could offer a rationale in the background for why you conducted this research.

Response: we have reviewed that manuscript and made necessary changes.

Read: The objective of NPPP, 2012 was “to put in place a regulatory framework for pricing of drugs so as to ensure availability of required medicines - ‘essential medicines’ – at reasonable prices even while providing sufficient opportunity for innovation and competition to support the growth of industry, thereby meeting the goals of employment and shared economic well-being of all.”

“Empirical literature suggests that while direct price control policies are effective in reducing prices and are able to control expenditures, they are unable to reduce medicine expenditures in the long run as manufacturers are able to find ways to increase sales of medicine formulations outside price control.

In the Indian context, the selective coverage of the policy led to concerns regarding the shift of sales from price controlled medicines to those outside price control but within the same class of medicines as a result of change in marketing priorities of the companies who would have an incentive to push medicines outside price control and the policy could lead to a shift toward other strengths of atorvastatin or other statins such as rosuvastatin.”

- Why did the authors select atorvastatin as their case study? The authors say that India has “the highest burden of cardiovascular diseases among developing nations”, but I thought they were interested in studying the general impact of the policy change on the market shares of medicines affected by the new regulation — rather than the specific impact on the market shares of cardiovascular products.

- o Are data available for other medicines? It would have been interesting to select a larger sample of widely-used medicines, or at least a few key products to improve the generalisability of the results.

Response: We have data available for other medicines as well. But pharmaceutical market, instead of being one single market, is a summation of a large number of sub-markets. The dynamics in each of these markets vary and therefore they cannot be expected to demonstrate the same impact from price regulation. For instance, statins market may have different impact from the regulation as compared to the market for oral anti-diabetics. Therefore, we studied the impact of price regulation on

statins market wherein only one statin Atorvastatin was under NLEM and therefore under price control. This explanation has been included in the main text as well.

Besides, the policy was implemented over a period of time with notification of price ceilings, was spaced out between 2013 and 2014, therefore the intervention time varied considerably. Prices of different strengths and dosage forms of the same medicine were also notified on different dates often in different months. This would make running a single interrupted time series model on multiple medicines at the same time complicated.

Keeping these things in mind, we chose the statins market as the burden on cardiovascular diseases in India is significantly high. Finally, it was the switch in utilisation as reflected in sales that was the outcome under study which required the use of relative share of price regulated medicine (atorvastatin) within the therapeutic segment (statins) under study.

The authors use many acronyms in the article, which makes it tedious to read. I would suggest that the authors remove all non-standard abbreviations which are not used at least four times in the paper and instead write out the full terms.

Response: Necessary modifications have been incorporated in the manuscript.

Materials and Methods — I have comments and questions about the methods.

- Which month in 2013 was the policy introduced? It appears to be June based on the first sentence under “statistical analysis”, but there needs to be a clear statement specifying when the policy came into effect.

Response: further clarifications have been included in the materials and method section of the manuscript.

- o Was this the only change that occurred around that time which might have influenced the market shares of atorvastatin? In the discussion (p. 11), you mention that “the Government of India has also implemented a host of policy intervention apart from NPPP 2012 in order to accelerate affordability of medicines and reduce out-of-pocket payments on medicines.” Could you please elaborate on this statement, since such policy interventions could threaten the internal validity of your findings?

Response: The other policies are related to government procurement and distribution of medicines across the different Indian states and therefore unlikely to have any impact on the private sector medicine utilization. We have further strengthened our model by introducing rosuvastatin, a statin outside price control to control for other interventions as well as time-varying confounders that might have affected the outcome of interest.

- In the appendix, could you please list the p-value for the test for positive and negative autocorrelation along with the Durbin-Watson test statistic? It would be good to include a statement that seasonality is unlikely to influence the outcome measure, if this is correct; otherwise you would need to account for seasonality in your model. [SEP]

Response: We have provided the p-values in the annexures. The following statement has been added in the text: “Seasonality is unlikely to influence the outcome measure as the group of medicines under study are intended for a chronic condition and meant to be consumed throughout the lifetime of the patients.

- Could you please re-run the model to allow for a lagged treatment effect? As shown in Figure 1, there are wild data points in the first five months after the policy change. You should drop these observations if you think the fluctuations occurred during a period of adjustment to the new policy and were not due to random variation, as done in other studies (Leopold et al 2014). Otherwise you might obtain an incorrect estimate of the treatment effect (Wagner et al 2002; Jandoc et al 2015). [SEP]

Response: We have incorporated an alternate model allowing for the implementation period. However, as expected, this did not considerably alter our results.

- Could you also re-run the model with total volume per month per capita as a secondary outcome measure, as done in similar studies (e.g., Leopold et al 2014; Garabedian et al 2012)? The impact of the policy change on market shares and volumes might differ.

Response: We ran the model and observed that the R square was extremely low indication that the model with per capita volumes was not a good fit.

We chose relative shares of sales as our dependent variable as it helped us study the switch between price regulated and unregulated medicine as well as implicitly control for the movements in the overall statins market. Descriptive statistics reported in the manuscript point out that the sales of statins had an upward trajectory in the study period. Taking shares instead of absolute volumes ensured that the

change in the level and trend of atorvastatin (5 mg and 10 mg) was not influenced by the overall trends in the statins market (that could be influenced by the changes in the disease burden, awareness levels, health-seeking behaviours etc.) and the changes could be attributed to the policy alone.

- To estimate the treatment effect relative to the control group, I would suggest you use the difference in the market shares between the two medicines over time as the outcome variable, as done in other studies (Law et al 2008, Penfold and Zhang 2008).

Response: we have made the suggested modification, although our results have not changed.

- Why did you select rosuvastatin instead of simvastatin (or another statin) as the control?

Response: The clarification has been provided in the methods section. Please see: rosuvastatin was chosen as control instead of other statins such as simvastatin, because rosuvastatin is the second highest selling plain formulation (i.e. non-fdc) in both value and volume terms after atorvastatin. We ran our model with simvastatin as control (dependent variable: difference between the market share of atorvastatin (5mg and 10mg) and simvastatin). The results are as follows:

Model 4

Variable	Coefficients	95% conf. interval		
Time	-0.27 (***)	-0.32	-0.21	
Intervention (level change)		-0.25	-0.91	0.41
Time after intervention (trend change)		0.10 (***)	0.04	0.16
_Const	32.53 (***)	31.97	33.10	
No. of observations	48 (pre-intervention: 17 and post-intervention: 31)			
R-squared	0.9655			

Note: *** refers p<0.001; ** refers to p<0.05

Our findings did not change significantly on using simvastatin instead of rosuvastatin as control.

- As mentioned above, it is unclear which medicines were subjected to the policy. Why was atorvastatin affected by the new regulation, but not other statins like rosuvastatin and simvastatin? This information is important for determining whether rosuvastatin is an appropriate control group or not.

Response: the explanation has been provided in the background section. Please see : The only statin under price regulation atorvastatin – 5mg and 10mg. The remaining strengths of atorvastatin such as 20mg and 40mg, other statins such as rosuvastatin and simvastatin as well as combinations containing atorvastatin such as atorvastatin+acetylsalicylic acid remain outside price regulation.

- The subsection on the null hypothesis is incorrect and misleading. The null hypotheses relate to each variable in your model, usually specifying that the value of a coefficient is 0, holding the other variables constant. As all the analyses were two-tailed with a type I error rate of 0.05, the model would pick up an increase or decrease in the market share of atorvastatin (not just a decrease, as you seem to suggest). Also, you are describing the alternative hypothesis in the text, not the null hypothesis.

Response: necessary modifications have been made in the text.

Results — I would suggest the authors rewrite this section after considering the comments above. Please ensure you follow the reporting recommendations of Jandoc et al (2015). I have a few minor points:

- Table 1 is difficult to read. I would indent the atorvastatin strengths to make it clear that those figures add up to the total across all strengths (first row). The authors may want to somehow distinguish the last two rows from the rest (e.g., add a blank row above), since they present different types of statistics. Also, the units in the last row are billions of standard units (not % market share), so the column headings do not apply.

Response: the table has been modified as follows for greater clarity:

	Sales volumes (%)				Sales values (%)				
	2012	2013	2014	2015	2012	2013	2014	2015	
(A) Atorvastatin (5mg & 10mg) (%)	32.34	28.87	26.29	24.1	25.84	22.4	17.01		
(B) Atorvastatin (Other Strengths) (%)	12.63	11.88	11.81	11.64	23.09	22.57	23.19		
(C=A+B) Atorvastatin (all strengths)	44.97	40.75	38.1	35.73	48.93	44.96	40.2		
	38.23								

(D) Rosuvastatin (all strength) (%)	15.79	17.53	19.15	20.12	19.65	22.66	25.16
	26.27						
(E=A-D) Difference between Atorvastatin (5mg & 10mg) & Rosuvastatin (%)	7.14	3.98	6.19	-0.26	-8.15	-10.97	
							16.55 11.34

• I would suggest you present the trends for both the treatment and control group (rosuvastatin market shares) in Figure 1, along with the differences between the two.

Response: Figure 2 provides the trends for both the treatment and control group as well as the difference.

• Could you please remove the sentence about the null hypothesis (p. 10) for the reasons mentioned above?

Response: suggested modifications have been made in the text.

Discussion — I have concerns about the generalisability of your findings. Also, without a clear statement of purpose in the background, it is difficult to understand the policy implications of your research, especially since you only examined the impact of the policy change on the market share of a single product.

• What can health stakeholders in India and elsewhere learn from this study? Why is this important evidence for the wider health policy audience? ^[1]_[SEP]

Response: We agree that we have evaluated the impact of price regulation on only one market; our findings are not representative for other medicines or formulation under price control. However, it should be noted that the impact of the policy could vary across therapeutic segments and medicines under consideration with the market dynamics in terms of the number of players, relative competition, the behaviour of the players in different segments varies, as does consumer behaviour, therefore there is merit in studying individual markets separately.

• A key limitation of this study is that the authors only analysed the market shares of atorvastatin, so it is difficult to generalise your results to other products (or even other strengths of atorvastatin than those included in your analysis). You may wish to discuss why these results may or may not hold for other products. ^[1]_[SEP]

Response: the following has been included in text: “However, it should be noted that the impact of the policy could vary across segments as the market dynamics in terms of the no. of players, relative competition, the behaviour of the players in different segments varies, as does consumer behaviour, therefore there is merit in studying individual markets separately.”

• Please say “possible reasons” instead of “notable reasons” (p. 10), since this is your own interpretation. ^[1]_[SEP]

Response: suggested modifications have been made in the text.

• Could you please discuss what your study adds to the literature? ^[1]_[SEP]

Response: Please see the following: “Empirical literature suggests that generally, direct price control policies are usually unable to reduce expenditures as manufacturers find ways to increase sales of medicines outside regulation. Recent research on price control of anti-hypertensive medicines from Korea reported some unintended effect of the policy i.e. drug price reduction resulted in drug overutilization and use of prohibited combinations. Also, the utilisation of drugs, which was not affected by price reduction, increased by 12.3%. The case of the statins market in India does not reflect this shift toward unregulated medicines.”

• Conclusion — I would rewrite this section to clearly and concisely communicate the key finding of your study and the wider policy implications of your results. I would not repeat points relating to the study sample, such as “Two strengths of Atorvastatin (20mg and 40mg) were included in ... left out of the list.” ^[1]_[SEP]

Response: This point provides additional information regarding the recent developments in the policy. We think it is important to provide this information to the readers and clearly mention that the impact of this development can be studied only after sufficient time has lapsed after implementation.

Minor comments [SEP]

- There are many grammatical mistakes and formatting errors in the paper, so please proof read the article again carefully. Here are some examples:
 - o “Regulation” is misspelled (keywords, p. 2) [SEP]
 - o “Another” should be capitalised (last sentence of paragraph 3, p. 4) [SEP]
 - o “Market-based pricing” should not be capitalised (1st paragraph p. 5)
 - o Reference 17 (p. 7) should appear in the same format as the others, i.e. superscript number
 - o The total sales volume was 28.9% not 28.8% (top of p. 8)
 - o There is an extra closing parenthesis (p.8) [SEP]
 - o You’ve already defined the acronym NLEM, so there is no need to re-define it in the conclusion (p. 12)

Response: suggested changes have been incorporated in the manuscript. [SEP]

- In the background section, the sentence “For example, the Organization for [SEP]Economic Cooperation for [sic] Development (OECD) countries require manufacturers to limit prices in exchange of [sic] subsidies they receive” (p. 4-5) is factually incorrect. You could say that “some” OECD countries do this, but it is not the case in all. You should also clarify what you mean by subsidies. [SEP]Please note that the name of the institution is the Organization for Economic Cooperation and Development (not “for”), and the sentence should read “for subsidies” (not “of subsidies”). [SEP]

Response: the section has been edited in the manuscript as follows: Some Organization of Economic Cooperation for Development (OECD) countries for instance, require manufacturers to limit prices in exchange for the subsidies they receive.

- I would not capitalize the generic names of medicines, like atorvastatin and rosuvastatin. [SEP]

Response: suggested change has been incorporated in the manuscript. [SEP]

- The anatomical therapeutic chemical classification system was developed by the WHO Collaborating Centre for Drug Statistics Methodology, not the European Pharmaceutical Market Research Association (p. 6), which is an industry association.

Response: both the EphMRA and the WHO have their own systems of ATC classification. Please refer to <http://www.ephmra.org/classification> for more information on the EphMRA system. IMS uses the EphMRA system of classification.

- The name of the organisation which provided the data is now QuintilesIMS, not IMS Quantile (p. 5)

Response: suggested change has been incorporated in the manuscript. [SEP]

- I would present the 95% confidence intervals for the point estimates in the abstract.

Response: suggested change has been incorporated in the manuscript. [SEP]

VERSION 2 – REVIEW

REVIEWER	Olivier J. Wouters Researcher, LSE Health, London School of Economics and Political Science, UK
REVIEW RETURNED	18-Jun-2018

GENERAL COMMENTS	BMJ Open (Manuscript #2018-024200) I appreciate that the authors have done their best to address the reviewer comments. The revised paper is more concisely and clearly written, and I now better understand the study objectives and methods. I still have concerns about the generalizability (i.e., external validity) of your findings. I realize that the statin market is an important one, and that statins are often used as “tracer” drugs in other studies of the impact of regulatory changes on drug prices, expenditures, or market shares. Yet I would find the study results and conclusions more compelling if you would present results for a
--

	larger sample of products than just a single cholesterol-reducing drug. Alongside the aggregate results, it would be useful to present findings for individual products, such as atorvastatin, to see if the results hold in individual therapeutic areas. As it stands, it is difficult to gauge whether your findings are generalizable to other medicines in India and, thus, whether the policy was effective or not. And, if the results are not generalizable to other parts of the Indian drug market, then this would make the study findings less relevant for policymakers and other health stakeholders, since presumably your interest lies in the impact of the policy itself than in atorvastatin market shares. And, finally, there remain a lot of grammatical mistakes and formatting errors in the paper, so please proof read the article again carefully. Here are some examples (n.b., this list is not exhaustive, so you will need review the entire paper):  • “drug price control” not “drug piece control”; “Pharmaceutical” instead of “Pharmaeetical” (both in the abstract, p.2) • In the abstract conclusions, it should say “did improve” (not “did improved”) • The name of the organization is “Organisation for Economic Co-operation and Development” (p. 3) • “segments” not “segmnets” (p. 8) • “rosuvastatin” not “rosuvatstatin” (p.14); also, Rosuvastatin is capitalized later on the page, but should be written with a lower-case r • Pre-intervention not pre-interventon (p.16) • Atorvastatin instead of atorvastattin (p.16) • Guidelines not guidlines (p. 17)
--	---

REVIEWER	Björn Wettermark Karolinska Institutet, Sweden
REVIEW RETURNED	23-Jul-2018

GENERAL COMMENTS	1) General comments Thank you for the opportunity to review this paper. It has previously been reviewed and the previous reviewers raised many important remarks around the background and rationale behind the study, clarity about the aims, justification behind the selection of drugs (statins) for inclusion, the selected outcomes measure (using shares and not absolute values), other potential interventions taking place during the study period, the statistical methods and implications for science and policy. I agree completely with the critical comments raised by the second reviewer in the first review. Most of them have been addressed in the manuscript which has improved substantially. However, I believe there is still room for improvement around some of the comments raised as well as some further issues discussed below. 2) Specific comments for revision: a) major Abstract: This summarises the findings well. However, I suggest to avoid statements on effects for results that are not significant. I also think the authors should carefully check when to write “sales volume” and “market shares” so it is clear which effect they refer to.
--

	I also think the conclusion in the abstract should be modified since the study has not shown whether access to medicines has improved. Background: This captures the key rationale behind the study. However, I lack some text on what is previously known from the literature on the impact of price regulations in other countries, and why (if there is any reason for it) India may be of specific interest. There are also some references missing for some important statements such as the sentence “Literature suggests that most industrial countries provide medicines free of charge for their citizens...”. Aims: Ok and corresponds well to the study undertaken. However, I suggest not to use the word “uptake” of medicines (e.g. background last sentence). Uptake more commonly relates to the introduction, while this study includes all utilization and not only incident patients. I would suggest to use the word “utilization” or “sales” instead. Methods, data: It is claimed that complete data are used, but later in the discussion it becomes obvious that access in the public sector cannot be studied, given the lack of national level public sector data? Perhaps this should be mentioned already here and its potential implications. Methods, statistical analyses: The section has become very long after all additions. Consequently, there is a risk that the readers may have difficulties to capture what has been done. If possible this should be organized more clearly with subheadings and bullets to facilitate for the readers. The rationale behind the selection of controls etc are thoroughly described. Part of this more relates to the discussion on strength and weaknesses with the method and could be moved to the discussion section. Results: The new tables add value since they present the total picture, but I lack some info on how the shares relate to the total volumes and values? And since monthly data is not presented it is difficult to tell whether the total changes are secular trends or not. I would suggest some text and data linking the annual data to the more detailed ITS: The discussion could further address the implications for science and policy. Clear suggestions were given by the previous second reviewer, and I believe they are not completely taken into account. b) minor Abbreviations used in tables should be explained in footers in connection to each table.
--	---

VERSION 2 – AUTHOR RESPONSE

Reviewer: 1

Reviewer Name: Olivier J. Wouters

Institution and Country: Researcher, LSE Health, London School of Economics and Political Science, UK

Competing Interests: None declared

I appreciate that the authors have done their best to address the reviewer comments. The revised paper is more concisely and clearly written, and I now better understand the study objectives and methods.

- I still have concerns about the generalizability (i.e., external validity) of your findings. I realize that the statin market is an important one, and that statins are often used as “tracer” drugs in other studies of the impact of regulatory changes on drug prices, expenditures, or market shares. Yet I would find the study results and conclusions more compelling if you would present results for a larger sample of products than just a single cholesterol-reducing drug. Alongside the aggregate results, it would be useful to present findings for individual products, such as atorvastatin, to see if the results hold in individual therapeutic areas.

As it stands, it is difficult to gauge whether your findings are generalizable to other medicines in India and, thus, whether the policy was effective or not. And, if the results are not generalizable to other parts of the Indian drug market, then this would make the study findings less relevant for policymakers and other health stakeholders, since presumably your interest lies in the impact of the policy itself than in atorvastatin market shares.

Response: The reviewers’ comments point out not the limitation of the study, but the scope. Our manuscript does not claim that our findings from the statins market are applicable to or generalizable across other therapeutic segments. Insofar as the scope of the study is well defined, we do not see a reason to dwell upon the question of generalizability. We had clarified in our previous responses to the reviewers why there is merit in studying different sub-markets in the pharmaceutical sector separately.

- And, finally, there remain a lot of grammatical mistakes and formatting errors in the paper, so please proof read the article again carefully. Here are some examples (n.b., this list is not exhaustive, so you will need review the entire paper):

- “drug price control” not “drug piece control”; “Pharmaceutical” instead of “Pharmaeutical” (both in the abstract, p.2)
- In the abstract conclusions, it should say “did improve” (not “did improved”)
- The name of the organization is “Organisation for Economic Co-operation and Development” (p. 3)
- “segments” not “segmnets” (p. 8)
- “rosuvastatin” not “rosuvatstatin” (p.14); also, Rosuvastatin is capitalized later on the page, but should be written with a lower-case r
- Pre-intervention not pre-interventon (p.16)
- Atorvastatin instead of atorvastattin (p.16)
- Guidelines not guidlines (p. 17)

Response: suggested modification has been incorporated in the manuscript at the appropriate places.

Reviewer: 2

Reviewer Name: Björn Wettermark

Institution and Country: Karolinska Institutet, Sweden

Competing Interests: None declared

1) General comments

Thank you for the opportunity to review this paper. It has previously been reviewed and the previous reviewers raised many important remarks around the background and rationale behind the study, clarity about the aims, justification behind the selection of drugs (statins) for inclusion, the selected outcomes measure (using shares and not absolute values), other potential interventions taking place during the study period, the statistical methods and implications for science and policy. I agree completely with the critical comments raised by the second reviewer in the first review. Most of them have been addressed in the manuscript which has improved substantially. However, I believe there is still room for improvement around some of the comments raised as well as some further issues discussed below.

2) Specific comments for revision:

a) major

Abstract: This summarises the findings well. However, I suggest to avoid statements on effects for results that are not significant. I also think the authors should carefully check when to write “sales volume” and “market shares” so it is clear which effect they refer to.

I also think the conclusion in the abstract should be modified since the study has not shown whether access to medicines has improved.

Response: suggested modification has been incorporated in the manuscript.

Background: This captures the key rationale behind the study. However, I lack some text on what is previously known from the literature on the impact of price regulations in other countries, and why (if there is any reason for it) India may be of specific interest. There are also some references missing for some important statements such as the sentence “Literature suggests that most industrial countries provide medicines free of charge for their citizens...”.

Response: previously known information from existing literature has been discussed in the discussion section in detail. We have excluded this from the background, to avoid repetition. Specifically, why India is of interest has been discussed in the background section wherein the problems with access to medicines in India and the rationale behind the price regulation policy have been discussed.

Aims: Ok and corresponds well to the study undertaken. However, I suggest not to use the word “uptake” of medicines (e.g. background last sentence). Uptake more commonly relates to the introduction, while this study includes all utilization and not only incident patients. I would suggest to use the word “utilization” or “sales” instead.

Response: suggested modification has been incorporated in the manuscript.

Methods, data: It is claimed that complete data are used, but later in the discussion it becomes obvious that access in the public sector cannot be studied, given the lack of national level public sector data? Perhaps this should be mentioned already here and its potential implications.

Response: suggested modification has been incorporated in the manuscript. The study analysed the impact of price regulation on atorvastatin in the private retail market. The price regulation policy does not impact public sector in anyway. The public sector purchases medicines through medical services corporations through tender based mechanism. Last bullet point in article summary captures this.

Methods, statistical analyses: The section has become very long after all additions. Consequently, there is a risk that the readers may have difficulties to capture what has been done. If possible this should be organized more clearly with subheadings and bullets to facilitate for the readers. The rationale behind the selection of controls etc are thoroughly described. Part of this more relates to the discussion on strength and weaknesses with the method and could be moved to the discussion section.

Response: suggested modification has been incorporated in the manuscript.

Results: The new tables add value since they present the total picture, but I lack some info on how the shares relate to the total volumes and values? And since monthly data is not presented it is difficult to tell whether the total changes are secular trends or not. I would suggest some text and data linking the annual data to the more detailed ITS:

The discussion could further address the implications for science and policy. Clear suggestions were given by the previous second reviewer, and I believe they are not completely taken into account.

Response: suggested modification has been incorporated in the manuscript. The section has been completely rewritten.

b) minor

Abbreviations used in tables should be explained in footers in connection to each table.

Response: suggested modification has been incorporated in the manuscript.

VERSION 3 – REVIEW

REVIEWER	Björn Wettermark Karolinska Institutet, Sweden
REVIEW RETURNED	16-Sep-2018
GENERAL COMMENTS	I believe the authors have sufficiently adressed the comments I raised in my previous review.